# Effect of the Surface Functionalization of Graphene and MWCNT on the Thermodynamic, Mechanical and Electrical Properties of the Graphene/MWCNT-PVDF Nanocomposites

**DOI:** 10.3390/polym14152976

**Published:** 2022-07-22

**Authors:** Mamdouh A. Al-Harthi, Manwar Hussain

**Affiliations:** 1Department of Chemical Engineering, King Fahd University of Petroleum and Minerals, Dhahran 31261, Saudi Arabia; 2Department of Materials and Chemical Engineering, Erica Campus, Hanyang University, Ansan 15588, Korea

**Keywords:** functionalized graphene, CNT, dynamic mechanical properties, PVDF, graphene, pressure sensor, nanocomposites, acid treatment, thermal mechanical properties, solution casting

## Abstract

The nanocomposites of poly(vinylidene fluoride) (PVDF) with pristine graphene nanoflakes (GNF) and a multi-wall carbon nanotube (MWCNT) were prepared by the solution casting method. Additionally, the GNF and MWCNT were functionalized by acid treatment, and nanocomposites of the acid-treated MWCNT/GNF and PVDF were prepared in the same method. The effect of the acid treatment of MWCNT and GNF on the mechanical, thermal and thermo-oxidative stability and the thermal conductivity of the MWCNT/GNF-PVDF nanocomposites was evaluated, and the results were compared with the untreated MWCNT/GNF-PVDF nanocomposites. In both cases, the amount of GNF and MWCNT was varied to observe and compare their thermal and mechanical properties. The functionalization of the GNF or MWCNT resulted in the change in the crystallization and melting behavior of the nanocomposites, as confirmed by the differential scanning calorimetry analysis. The addition of the functionalized GNF/MWCNT led to the improved thermal stability of the PVDF nanocomposites compared to that of the non-functionalized GNF/MWCNT-PVDF nanocomposites. The thermal and electrical conductivity of the functionalized and non-functionalized GNF/MWCNT-PVDF composites were also measured and compared. The functional groups, crystal structure, microstructure and morphology of the nanocomposites were characterized by Fourier transformed infrared spectroscopy (FTIR), X-ray diffraction (XRD) and scanning electron microscopy (SEM), respectively.

## 1. Introduction

The addition of various non-functionalized and functionalized fillers to fabricate polymer nanocomposites is a relatively new concept to improve the physical, mechanical and thermal properties of the polymer. The fabrication of polymer nanocomposites using rigid fillers such as SiO_2_, TiO_2_, carbon black and carbon nanotubes (CNTs) has drawn much attention in recent years because the composites show significantly improved mechanical, thermal, structural and electrical properties compared to those of the pristine material due to the dispersion of inorganic nanoparticles into a polymeric matrix giving rise to a hybrid inorganic–organic system where a number of different interactions are established, which strongly affect the physicochemical properties of the host matrix [1,2,3,4,5,6,7]. Recently, graphene has attracted much interest owing to its excellent thermal and electrical properties and vast application prospects.

Poly(vinyl difluoride) (PVDF) has been extensively utilized in different research and application areas because of its excellent stability, high-temperature tolerance and oxidation resistance/stability [8,9]. Moreover, PVDF shows excellent melt-mixing processibility to prepare various composites and thin membranes. Moreover, PVDF is soluble in many solvents such as dimethyl formamide (DMF) and N-methyl-2-pyrrolidone (NMP), which facilitates the preparation of filler-dispersed nanocomposites by the solution casting method.

To achieve the improved properties of polymeric materials by dispersing CNT or graphene, a significant surface/interface interaction between the fillers and the matrix is essential. However, such strong interfacial bonding between the dispersed nanomaterials is extremely difficult to achieve in the absence of functional groups on their surfaces. Furthermore, owing to the large surface-to-volume ratio and strong Van der Waals interactions, the nanoparticles agglomerate in the polymer matrix. Therefore, the interfacial adhesion between the nanoparticles and the polymeric matrix needs to be optimized. To increase the interfacial adhesion between the polymer and nanofillers, the surface modification of the nanofillers (CNTs or graphene) has been explored. The noncovalent functionalization, which primarily involves Van der Waals force, does not affect the structure of the nanotubes [10]. In contrast, the covalent interaction greatly enhances the nanofiller dispersibility in the polymer matrix owing to the defects on the nanotubes’ surface. The most common approach for the covalent functionalization is the treatment with various inorganic acids using high-power sonication [11]. These oxidative treatments usually result in the formation of various surface functional reactive groups, such as hydroxyl, carbonyl and carboxylic acid [12]. Presently, many researchers explore chemical oxidation by acid treatment to improve the interfacial bonding between CNT and Graphene [13,14,15,16]. The functionalization of CNT or graphene with carboxyl (–COOH), hydroxyl (–OH) and carbonyl groups (–C=0) is generally performed using sulfuric and nitric acid [17,18]. Shanmugharaj et al. reported the presence of –COOH, –OH and –C=O functional groups by using potassium dichromate with H_2_SO_4_ [19].

In our previous work, we studied the thermo-mechanical dynamic properties of unmodified CNT and graphene-dispersed PVDF and evaluated and compared their properties. In the present work, we modified the multiwall carbon nanotube (MWCNT) and graphene nanoflakes (GNF), introducing surface functional carboxylic groups by acid treatment and then dispersing them in the PVDF polymer matrix. We studied the dynamic, mechanical, and thermal properties primarily to evaluate the thermo-oxidative stability of the PVDF composites. Further, we compared the results obtained with functionalized and non-functionalized MWCNT/GNF-PVDF nanocomposites.

## 2. Materials

PVDF (MW~530,000 g/mol) and N, N-Dimethylformamide (DMF) were purchased from Sigma Aldrich and used without any further purification. GNF and MWCNT were purchased from Grafen Chemical Industries, Turkey and Nanocyl, Belgium, respectively, and used without any further treatment and after acid treatment in different reactions. Figure 1 shows the TEM images of MWCNT and graphene before acid treatment.

## 3. Experimental

### 3.1. Acid Treatment of MWCNT and GNF

The acid treatments of MWCNT and GNF were performed separately. Pristine MWCNT/GNF (2 g) was added to a mixture of sulfuric and nitric acid (3:2 *v*/*v* ratio). The beaker containing the resulting mixture was then sealed and stirred for 24 h at 60 °C. After 24 h, the mixture containing MWCNT/GNF was filtered using filter papers and washed thoroughly with deionized water until the pH of the mixture was 7. The filtered MWCNT/GNF was then washed with acetone to remove the impurities of nanotube sidewalls. Finally, the acid-treated MWCNT/GNF was dried at 60 °C in a vacuum oven for 24 h. A schematic of the functionalization reaction of MWCNT/GNF is shown in Figure 2.

### 3.2. Fabrication of GNF/MWCNT-PVDF Nanocomposites

GNF/MWCNT-PVDF nanocomposites were prepared using the solution casting method. During the preparation of the composites, a 5 wt.% of MWCNT/GNF with respect to PVDF was taken. First, the required amount of PVDF was dissolved in 50 mL of DMF (N, N-Dimethyl formamide) at 90 °C under constant stirring for 4 h. Then, the PVDF solution was mixed with a previously prepared stable dispersion of MWCNT/GNF in 40 mL DMF by ultra-sonication with an amplitude of 500 W for 10 min. The solution mixture was stirred for another 1 h using a magnetic stirrer at a rate of 400 rpm. Then, the mixture was degassed for 10 min in a vacuum oven and poured into a stainless sheet petri dish placed on a leveled flat surface. The sample was then allowed to dry at 50 °C for 3–4 days. Finally, the dried films were peeled off carefully. Figure 3 shows a schematic representation of the preparation of the nanocomposites. The nomenclatures of the prepared nanocomposites are shown in Table 1.

## 4. Characterization Techniques

### 4.1. Thermogravimetric Analysis (TGA)

TGA analysis was performed using a Thermogravimetric analyzer (SDT Q600, TA Instruments). A portion of 6 mg of the samples was heated in the temperature range of 30–850 °C at a heating rate of 10 °C/min under a nitrogen atmosphere with a purge flow of 100 mL/min.

### 4.2. Differential Scanning Calorimetry (DSC)

DSC analysis of the samples was performed at a heating and cooling rate of 10 °C/min in the temperature range of −80 to 250 °C under a nitrogen atmosphere using a TA instrument DSC Q1000 differential scanning calorimeter. A portion of 6 mg of the samples was used for DSC analysis. The melting and crystallization behaviors of the samples were calculated by DSC.

### 4.3. Dynamic Mechanical Analysis (DMA)

Discovery DMA 850, manufactured by TA Instruments, was utilized for the DMA analysis. The DMA and properties of the samples were determined in a temperature range of −80 to 130 °C using the oscillation temperature ramp with a constant frequency of 1 Hz. Rectangular samples were taken from the composite sheet with an average dimension of 17.5 mm × 13 mm × 0.6 mm for the measurements.

### 4.4. Thermal Conductivity Measurements

The thermal conductivity of the composites was measured using a TA Instruments Fox 50 Heat Flow Meter. Circular samples with a diameter of 2 inches were taken from the sheet for the thermal conductivity measurements.

### 4.5. Electrical Transport Measurements

The electrical conductivity was measured using an Ossila Four-Probe system at ambient temperature. The dimension of the samples for the electrical transport measurements was 18 mm × 13 mm × 0.5 mm (length × width × thickness). The samples were cleaned using ethanol before measurements to avoid the influence of dust on the electrical resistivity of the sample.

### 4.6. Fourier Transformed Infrared Spectroscopy (FT-IR)

FTIR spectra of the treated MWCNT and GNF were analyzed using a Perkin Elmer Spectrum 1000 FTIR spectrometer.

### 4.7. Scanning Electron Microscopy (SEM)

The surface morphology of the MWCNT/GNF-PVDF composites was analyzed by a scanning electron microscope (SEM, FEI Quanta FEG 250). The PVDF sheet samples were gold coated (10 nm) using a Quorum Q150R-S sputter coater unit for the SEM analysis.

## 5. Results and Discussion

Effect of Acid Treatment of MWCNT and GNF

The FTIR spectra of pristine and acid-treated MWCNT and GNF are shown in Figure 4a and Figure 4b, respectively. The carbon molecules absorb the incidental IR radiation and generate signals at the stretching and vibration frequencies of the atoms [20]. Figure 4a shows the comparative FTIR spectra of pristine MWCNT and acid-functionalized MWCNT. Peaks around 2800–3900 cm^−1^ are attributed to the characteristic stretching frequencies of C–H and –OH bonds, which indicate the presence of hydroxyl and carboxylic groups, respectively. The appearance of wide peaks at 3888 and 3629 cm^−1^ in acid-treated MWCNT indicates the presence of –OH and –COOH functional groups, respectively. Another peak at 1556 cm^−1^ corresponding to the C=O stretching appears after acid treatment, which indicates the presence of carboxylic groups due to the surface oxidation [21]. Similar peaks are observed in the acid-treated GNF in Figure 4b. The presence of wide peaks at 3895 and 3609 cm^−1^ in the acid-treated GNF is indicative of –OH and –COOH functional groups, respectively. Peaks at 650 cm^−1^ indicate the bending vibration of C–H bonding [22]. The narrow peak at 1665 cm^−1^ is attributed to the C=O stretching, which indicates the presence of carboxylic groups due to the surface oxidation.

Figure 5 shows the overlay of the FTIR spectra of all MWCNT/GNF-PVDF nanocomposites. The characteristic peak of carbonyl groups at 1736 cm^−1^ is observed (Figure 5) in the FTIR spectra of the nanocomposites. This peak is attributed to the C=O stretching mode of carboxylic acids [23]. The presence of the peaks in the wavenumber ranges of 2884–2978 cm^−1^ and 3600–3888 cm^−1^ in Figure 5 is due to the formation of –COOH (carboxylic acids) on the surfaces of MWCNT and GNF. Due to the oxidative treatment with strong acids, defects and smaller fragments were observed in nanotubes. Similar behavior was observed by many researchers previously [24,25,26]

The crystallinity and thermodynamic behavior of the acid-treated MWCNT/GNF-PVDF composites prepared by the solution casting method were analyzed by DSC (Figure 6). The thermodynamic properties, including the crystalline temperature (T_c_), melting temperature ™, crystallinity (%) and heat of fusion (H_f_), are summarized in Table 2. It is well known that the mechanical properties of semi-crystalline polymers depend on their crystallinity and internal microstructure [27].

It is evident from the DSC curves (Figure 6a,b) that the acid treatment of graphene or MWCNT affects the crystallization temperature, melting temperature and crystallinity of the nanocomposites. The crystallization onset (To) and crystallization peak of the unmodified and acid-modified MWCNT-PVDF composites were found at 140.04 °C and 141.64 °C, respectively. The melting peak shifted to 160.37 °C from 159.9 °C after the acid treatment. The crystallization peak of the acid-treated MWCNT-PVDF samples shows a crystallinity of 41.22%, which is 3–4% higher than that of the untreated MWCNT (37.18%). The introduction of MWCNT into polymeric materials leads to an increase in T_c_ because MWCNT acts as a nucleating agent, which promotes the faster growth of PVDF crystals [28,29]. The further increase in crystallinity for the treated MWCNT promotes nucleation efficiency by reducing the polymer mobility and converting PVDF from the thermal type to the athermal type in the nanocomposites [28]. Similarly, the T_o_ and crystallization peaks of the unmodified and acid-modified GNF-PVDF composites were found at 133.04 °C and 134.97 °C, respectively. The melting peak shifted slightly from 159.63 °C to 159.43 °C after acid treatment, indicating the reduction in polymer mobility and chain flexibility [29]. The peak crystallization temperature of the untreated GNF-PVDF sample was found to be 133.04 °C; however, it was decreased to 134.97 °C after acid treatment, which can be attributed to the decrease in entropy change during the melting process in the presence of nanofillers [30,31]. The degree of crystallinity (X_c_) of the untreated GNF-PVDF film was found to be approximately 49.19%, which slightly increased to 49.79% for the acid-treated GNF-PVDF nanocomposites.

The addition of rigid fillers increases the thermal stability of a polymeric material. The improvement in the thermal stability of graphene and CNT-based nanocomposites has been reported in the literature. It has been reported that the addition of graphene/CNT increases the thermal stability of various polymers such as PMMA, PS and PVDF [32,33,34,35,36]. Figure 7a,b show the TGA and DTA analysis, respectively, of the unmodified and acid-modified GNF-PVDF composites. Figure 8a,b show the TGA and DTA graphs, respectively, of the unmodified and acid-treated MWCNT-PVDF.

The TGA curves of both the unmodified and acid-modified GNF-PVDF nanocomposites showed good thermal stability with no significant mass change up to 400 °C. The initial onset degradation (T_degrad onset_) started at approximately 458 °C and 425 °C for the unmodified GNF composites and virgin PVDF, respectively. For the modified GNF-PVDF, T_degrad onset_ and the degradation peak (T_degrad Peak_) appeared at 461 °C and 476 °C, respectively. The values of the T_degrad onset_ and T_degrad Peak_ for the different composite samples are shown in Table 3. Figure 8a shows that the unmodified MWCNT nanocomposites exhibit the highest T_degrad Peak_ at 448 °C, followed by the acid-modified MWCNT-PVDF composites at 471 °C. This might be due to the presence of inhomogeneous MWCNT in the PVDF matrix in the test sample. The amount of the nanocomposite residue after the TGA analysis increased with the addition of GNF/MWCNT and increased further for the acid-modified GNF/MWCNT-PVDF nanocomposites. Therefore, we observed an improvement in the thermal stability for both of the cases, which also results in the formation of protective layers (char formation) in the PVDF matrix during the melting process [37]. The thermal stability of the GNF-PVDF nanocomposites showed less of an improvement than that of the MWCNT-PVDF nanocomposites, which might be associated with the two-dimensional planer structure of the GNF. This might occur owing to the nano-confinement, as explained by Chen et al. [38,39].

To understand the dispersibility of MWCNT/GNF in the PVDF matrix, the thermal conductivity of the composite samples was investigated at different temperatures, and the results are displayed in Table 4. The addition of GNF or MWCNT significantly increased the thermal conductivity of the polymer matrix at room temperature. This trend is similar to the previously reported observations [40,41]. Researchers observed a significant enhancement in the thermal conductivity at a low graphene loading. The thermal conductivity of the untreated GNF composites was found to be approximately in the range of 0.17–0.19 W/mK. In contrast, the modified GNF-PVDF nanocomposites showed a thermal conductivity of 0.265 W/mK, which is approximately 45% higher than that of the unmodified GNF-PVDF nanocomposites. Similarly, the unmodified MWCNT composites showed a thermal conductivity range of 0.166–0.17 W/mK. The modified MWCNT-PVDF composites showed a thermal conductivity of 0.28 W/mK, which is 55% higher than that of the unmodified MWCNT composites. A few research groups explained the increase in the thermal conductivity of the PVDF composites after the addition of nanofillers; the nanofillers make a bridge between the PVDF spherulites and result in an enhanced heat transfer between the spherulites. As PVDF is a semi-crystalline polymer, the boundary between the amorphous regions and semi-crystalline regions shows a thermal interface resistance.

Table 5 shows the electrical conductivity of all the prepared nanocomposites. The PVDV matrix exhibited insulating behavior and did not show any electrical conductivity. However, by the addition of 5% GNF, it showed an electrical conductivity of 18.29 S/m, which was further dramatically enhanced (100%) to 35.65 S/m by the addition of modified GNF. The increase in the electrical conductivity of insulating polymers by the addition of electro-conductive GNF has been previously reported. However, a 100% improvement in the electrical conductivity by the acid modification of GNF has not been reported earlier. We believe this large enhancement occurred owing to a better dispersion of the acid-modified GNF in the PVDF matrix; the planer nature of GNF and the presence of functional groups on the GNF surfaces facilitate the jumping of ions, which can be explained by the tunneling effect. The MWCNT-PVDF composites showed an electrical conductivity of 26.82 S/m, which is significantly higher than that of pristine PVDF polymer. However, the acid-modified MWCNT-PVDF composite sample did not exhibit further enhanced electrical conductivity. This might be due to the low dispersibility of MWCNT in the PVDF matrix, defects on the MWCNT surface and the decrease in the nanotube length by the acid treatment [42].

Figure 9 shows the loss modulus of the treated and untreated MWCNT/CNF-PVDF nanocomposites prepared by the solution casting method as a function of temperature (−80 °C to +80 °C). The loss modulus of both the GNF and MWCNT-PVDF composites increased when the GNF and MWCNT were functionalized by acid treatment. The glass transition temperatures (T_g_) of both the composites were clearly observed. The T_g_ peak was observed at approximately −40 °C and −42 °C for the untreated and treated MWCNT-PVDF composites, respectively. The GNF-PVDF composites showed higher loss modulus values compared to the MWCNT-PVDF composites. This is due to the planer shape of graphene, which can absorb the higher load during the loss modulus measurement. The T_g_ peak was observed at approximately −35 °C and −38 °C for the untreated and treated GNF-PVDF composites, respectively. The loss modulus peak indicates interactions between the filler and matrix. It is also associated with the changes in the internal frictions between the filler and the matrix, the molecular motions, the morphology and the dispersion of the filler in the matrix. A single and strong peak confirms good interfacial interactions and no phase separation. The decrease in the loss modulus after the peak value is attributed to the free movement of the polymer chain present in the system [43]. The beta and gamma relaxation of the untreated MWCNT-PVDF polymer composites were observed at 20 °C and 60 °C, respectively, which indicated polymer mobility even after the addition of MWCNT nanofillers. In contrast, the treated MWCNT-PVDF nanocomposites showed slightly lower beta and gamma relaxation peak values, indicating the stiffness of the composites caused by the acid treatment, which reduced the mobility of the polymer chains. The untreated and treated CNF-PVDF composites showed a significantly better result than the MWCNT-PVDF nanocomposites. The beta relaxation peak was wider and appeared at a slightly lower temperature, suggesting more flexible and toughened nanocomposites.

Dynamic mechanical analysis was carried out to determine the various dynamic mechanical properties such as the storage modulus (E’), loss modulus (E″) and damping coefficient (tan δ) of the prepared samples as a function of temperature 80 °C to +80 °C). Dynamic mechanical properties were measured to confirm the dispersibility of MWCNT/GNF and the relative interactions between the PVDF polymer and the fillers. Figure 10 shows the storage modulus of the treated and untreated MWCNT/CNF-PVDF nanocomposites.

From the figure, two phenomena were observed. Firstly, the storage moduli values of the treated MWCNT-PVDF composites were significantly higher than those of the untreated MWCNT-PVDF composites, which suggested that the functionalization of MWCNT improved the dispersity of the fillers and the polymer–filler interaction. Secondly, the storage moduli of the values of the treated and untreated GNF-PVDF composites were not significantly different. Nanocomposites prepared by the solution mixing technique generally result in the better dispersion of the nanofillers in the matrix and good interfacial bonding between the polymer and nanofillers. The storage moduli of the GNF-PVDF nanocomposites increased with the increasing GNF content because of the polymer–GNF interactions; a decrease in the polymer mobility due to the presence of GNF resulted in the increased viscoelasticity and stiffness of the composites. GNFs separate the polymer chains and hinder the polymer mobility, thus increasing the reinforcement during the load transfer, which enhances the storage modulus. The similar storage modulus of the treated and untreated GNF-PVDF nanocomposites can be explained as due to GNF’s high aspect ratio, very high specific surface area and high surface energy, which restrict the polymer mobility and better dispersion. [44]. The decrease in the storage modulus with increasing temperatures is attributed to the energy dissipation due to the cooperative motion of the polymer chains [45].

Figure 11 shows the delta T of the treated and untreated MWCNT/CNF-PVDF nanocomposites as a function of temperature. The delta T indicates the T_g_ of the composites. The T_g_ of a polymer can be changed by the addition of rigid filler particles. In the present study, the T_g_ of the GNF-PVDF composites changed owing to (i) the decrease in the mobility of polymer chains by the addition of GNF and (ii) the restriction of segmental motion by graphene layers [46].

The T_g_ of the nanocomposites primarily depends on the dispersion of the filler particles. The sharp T_g_ of the melt-mixing nanocomposites indicates better nanofiller dispersion and nanofiller–polymer interaction in the PVDF matrix. A sharp T_g_ peak was observed at a lower temperature of −38 °C, and a wider and separate gamma relation behavior was observed at a higher temperature for both the treated and untreated GNF-PVDF nanocomposites. A wide and initially low T_g_ was observed for the MWCNT-PVDF nanocomposites. The T_g_ became gradually wider with the increasing measuring temperature. At higher graphene contents, the aggregation and poor dispersion of graphene, resulting in a lower and wider T_g_ peak, were reported previously; however, this was not the case in our present study [47,48]. The inhomogeneous graphene distribution in the polymer matrix was also found to affect the relaxation behavior [49].

Figure 12 shows the XRD patterns of the unmodified and modified MWCNT/GNF-PVDF nanocomposites. The intensities of the XRD peaks at 2θ values of 21° and 27.5° were found to be higher for the modified GNF-PVDF nanocomposites compared to those for the unmodified nanocomposites. Further, a broadening of the peak at 2θ = 40° was observed for the MWCNT/GNF-PVDF nanocomposites. A new peak at 2θ = 56° appeared for the modified MWCNT/GNF-PVDF nanocomposites. The broadening of the graphite peak suggests a decrease in the crystallite size of graphite after the oxidation, whereas an increase in the intensity of the graphite peak for the oxidized samples confirms an increase in the graphite component in the samples due to the dissolution of amorphous carbon in acids [50].

The XRD peaks at 2θ = 21 and 27.5° and the broadening of the peak at 2θ = 40° can be observed in both the unmodified and modified MWCNT-PVDF composites (Figure 13). A new peak at 12° was observed. The treatment of CNT with acids leads to the dissolution of weakly condensed carbon and the partial removal of the CNT growth catalyst from the cavities of nanotubes. Tsang et al. reported that the acid treatment resulted in the opening of the CNT edges [51].

Microstructural analysis of the nanocomposites was carried out using SEM. In the SEM images (Figure 14), MWCNT and GNF were not visible. However, the microstructure of the PVDF molecules was observed in the SEM images. In the SEM image of the MWCNT-PVDF nanocomposites (Figure 14b), PVDF molecules were found to be homogeneously dispersed. Further, a tougher bonding with a lesser number of voids indicates a toughened polymer. The SEM image of the GNF-PVDF nanocomposites (Figure 14b) shows that the PVDF molecules are tightly bonded; however, a large number of voids are observed, which result in a wide T_g_, delta T and other thermodynamic mechanical properties as compared to those of the MWCNT-PVDF nanocomposites.

The pristine PVDF shrinks after drying. However, in the case of the filler-added PVDF, the shrinkage is reduced. Because of the shrinkage, the pure PVDF sample does not exhibit good mechanical properties.

## 6. Conclusions

GNF- or MWCNT-dispersed PVDF nanocomposites were successfully fabricated by the solution casting method. GNF and MWCNT were treated with strong acids for surface functionalization. The thermo-mechanical properties, thermal properties and thermal conductivities of the nanocomposites were investigated, and it was found that the acid-treated GNF/MWCNT composites show better performance compared to the untreated GNF/MWCNT composites. The results were supported by DSC, TGA, DTG and XRD investigations. The thermal properties obtained from the TGA and DMA analysis confirmed the above findings and indicated that the acid treatment of GNF and MWCNT significantly improved the thermo-mechanical and thermal properties of the composites.

## Figures and Tables

**Figure 1 polymers-14-02976-f001:**
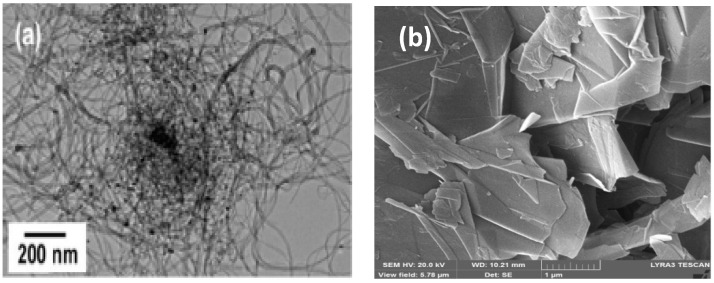
TEM images of (**a**) MWCNT and (**b**) GNF without treatment.

**Figure 2 polymers-14-02976-f002:**
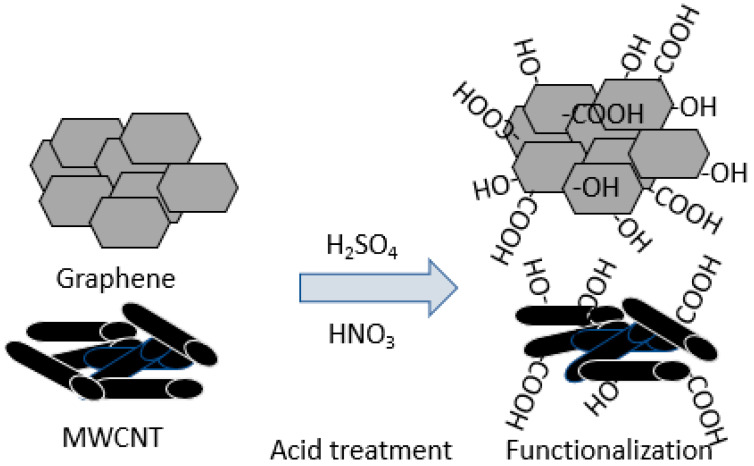
Schematic representation of the functionalization reaction of GNF and MWCNT.

**Figure 3 polymers-14-02976-f003:**
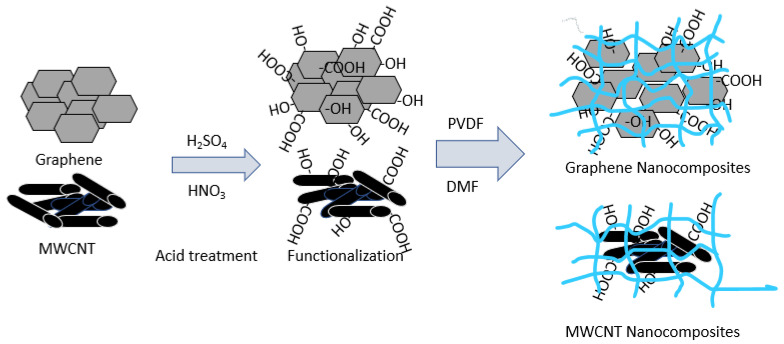
Schematic representation of the preparation of MWCNT/GNF-PVDF nanocomposites.

**Figure 4 polymers-14-02976-f004:**
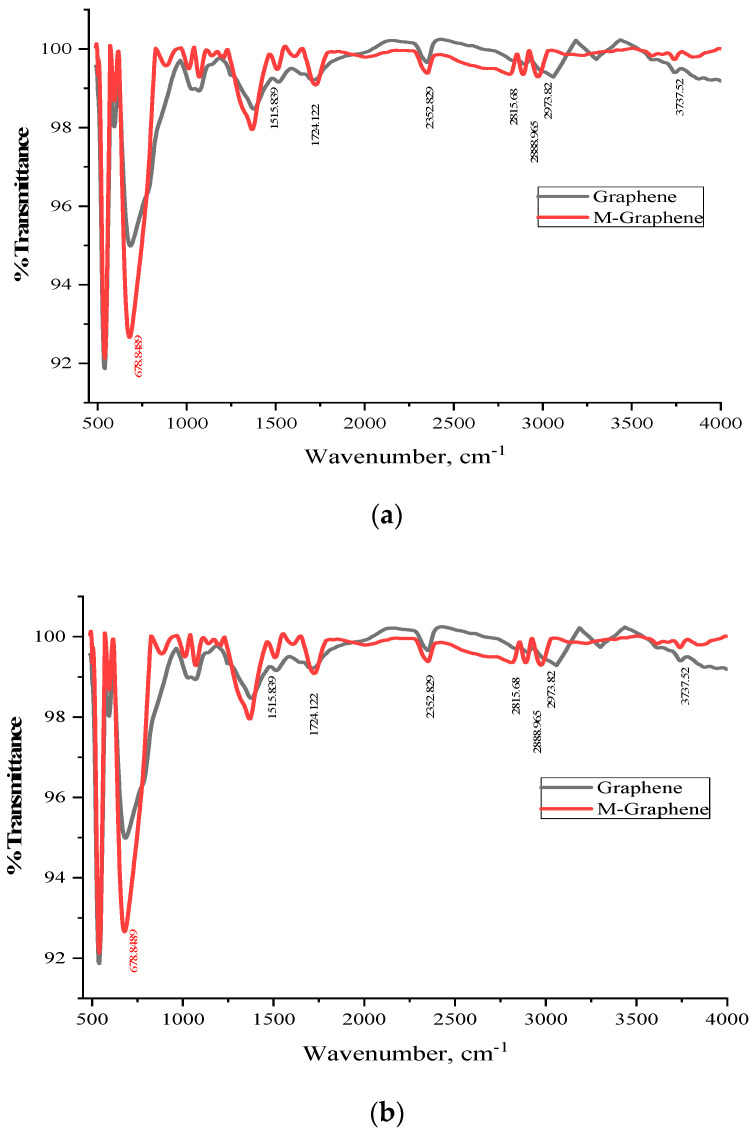
FTIR spectra of pristine and acid−treated (**a**) MWCNT and (**b**) GNF.

**Figure 5 polymers-14-02976-f005:**
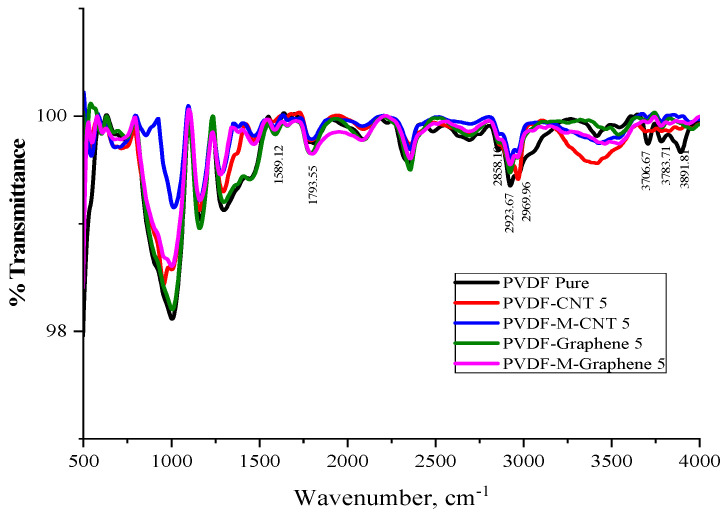
Overlay FTIR spectra of the acid-treated MWCNT/GNF-PVDF nanocomposites.

**Figure 6 polymers-14-02976-f006:**
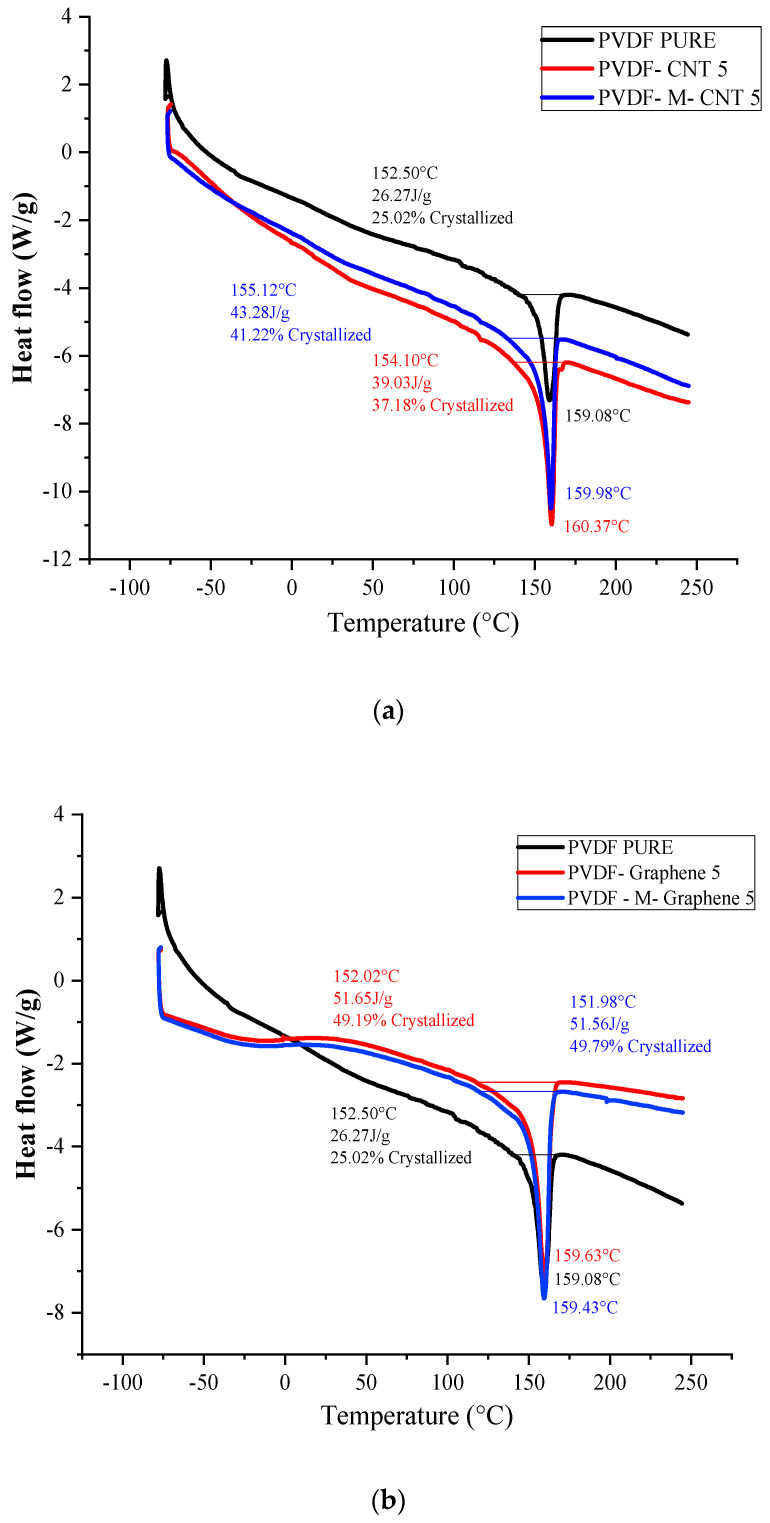
(**a**) DSC graphs of the MWCNT-PVDF and (**b**) GNF-PVDF nanocomposites.

**Figure 7 polymers-14-02976-f007:**
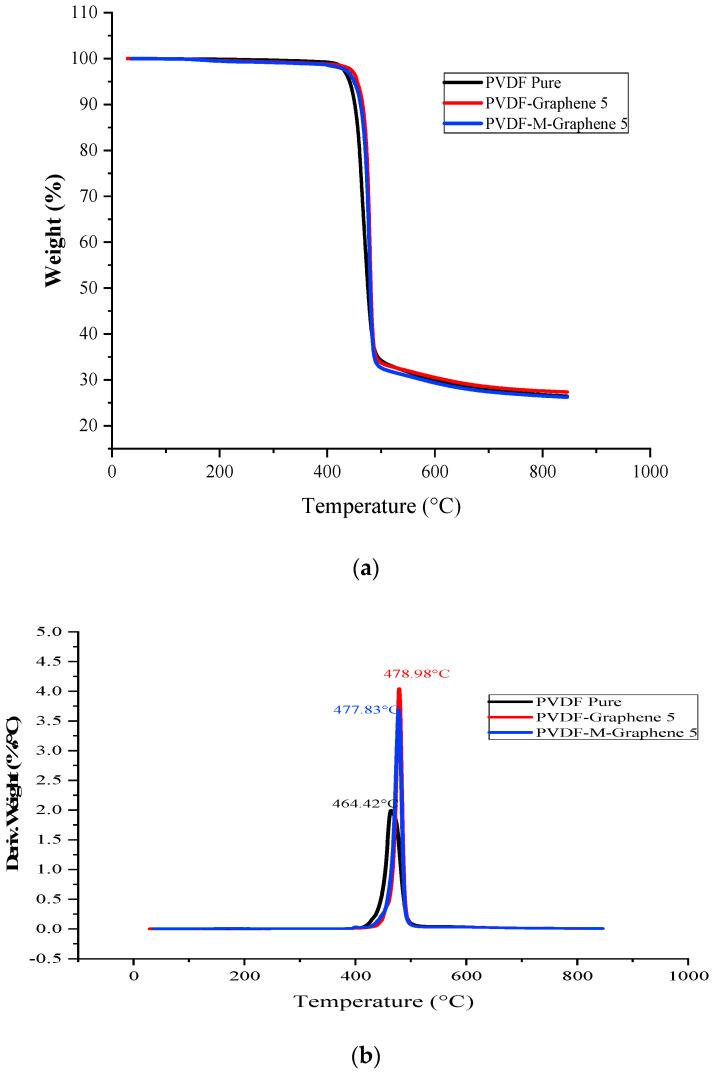
(**a**) TGA graphs and (**b**) DTG graphs of the GNF−PVDF nanocomposites.

**Figure 8 polymers-14-02976-f008:**
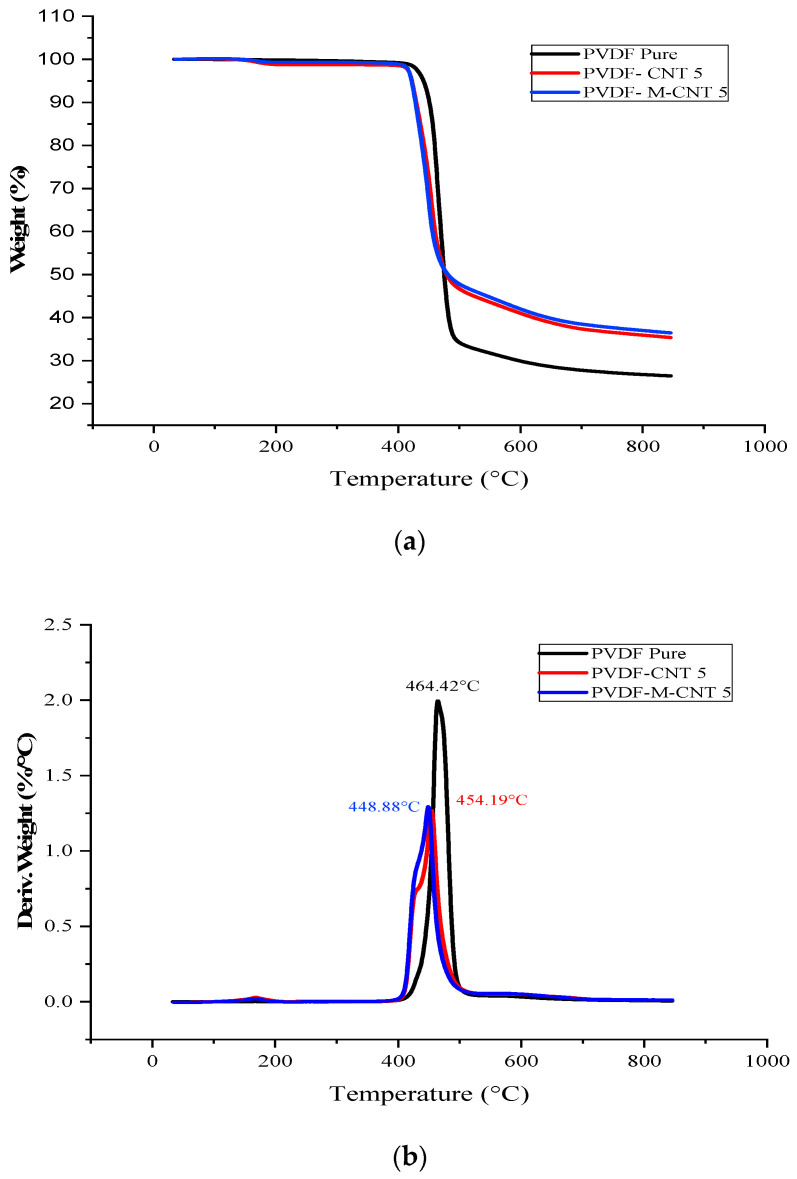
(**a**) TGA graphs and (**b**) DTG graphs of the MWCNT-PVDF nanocomposites.

**Figure 9 polymers-14-02976-f009:**
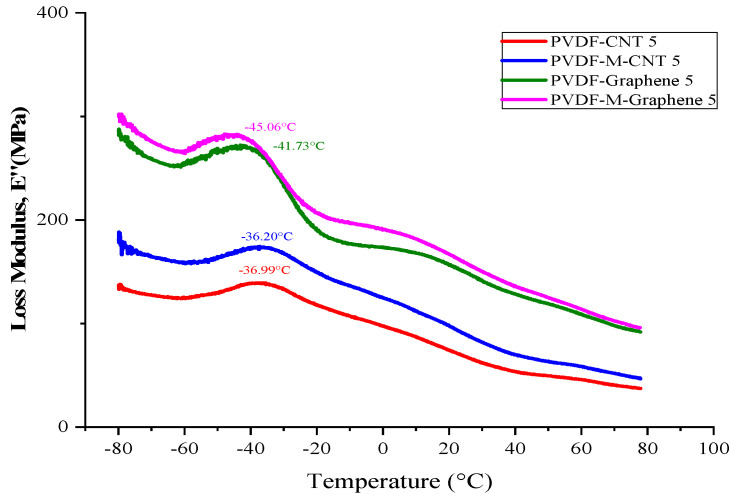
Loss modulus of the acid-treated and untreated MWCNT/GNF−PVDF nanocomposites.

**Figure 10 polymers-14-02976-f010:**
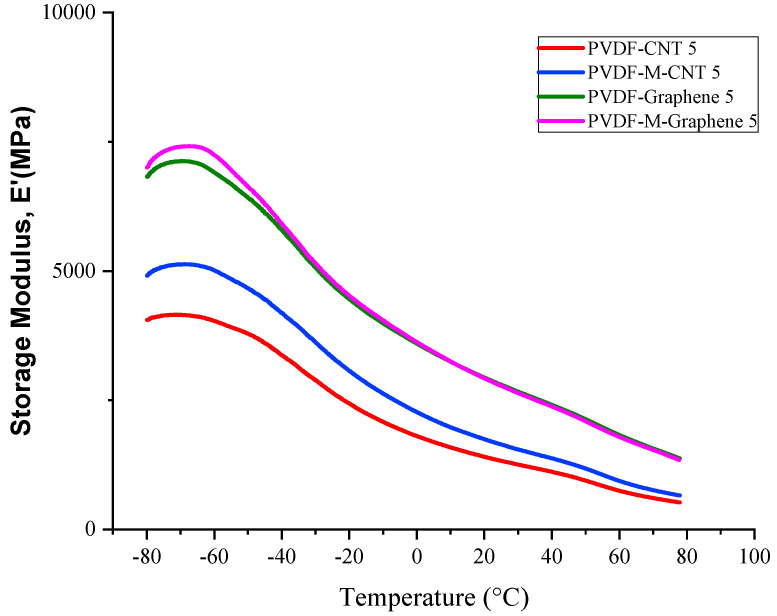
Storage modulus of the treated and untreated MWCNT/CNF-PVDF nanocomposites.

**Figure 11 polymers-14-02976-f011:**
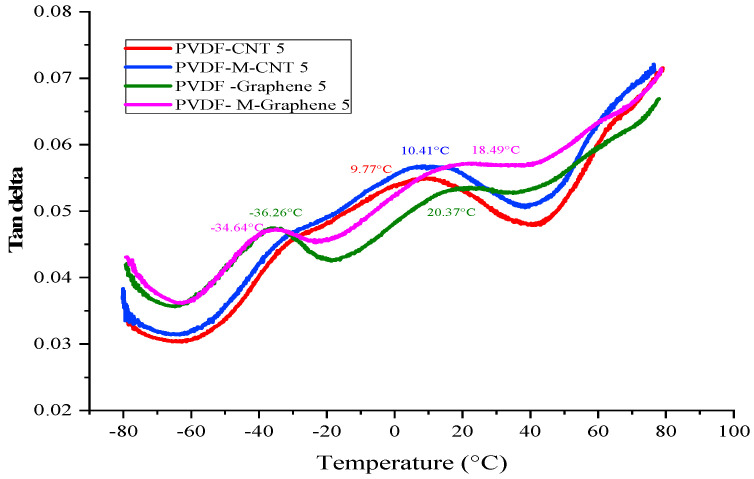
Delta T of the treated and untreated MWCNT/CNF-PVDF nanocomposites.

**Figure 12 polymers-14-02976-f012:**
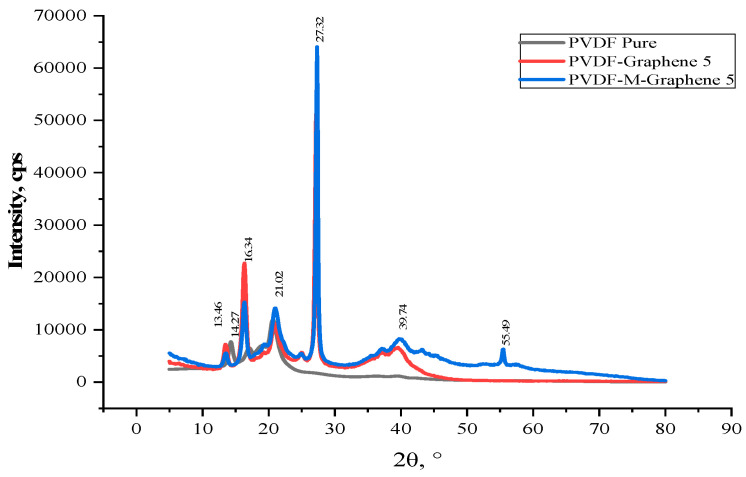
XRD patterns of PVDF and the unmodified and modified GNF-PVDF nanocomposites.

**Figure 13 polymers-14-02976-f013:**
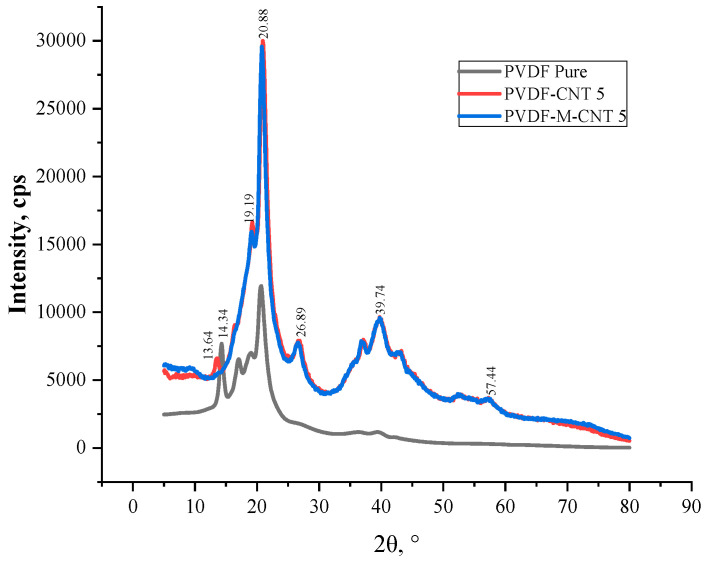
XRD patterns of PVDF and the unmodified and modified GNF-PVDF nanocomposites.

**Figure 14 polymers-14-02976-f014:**
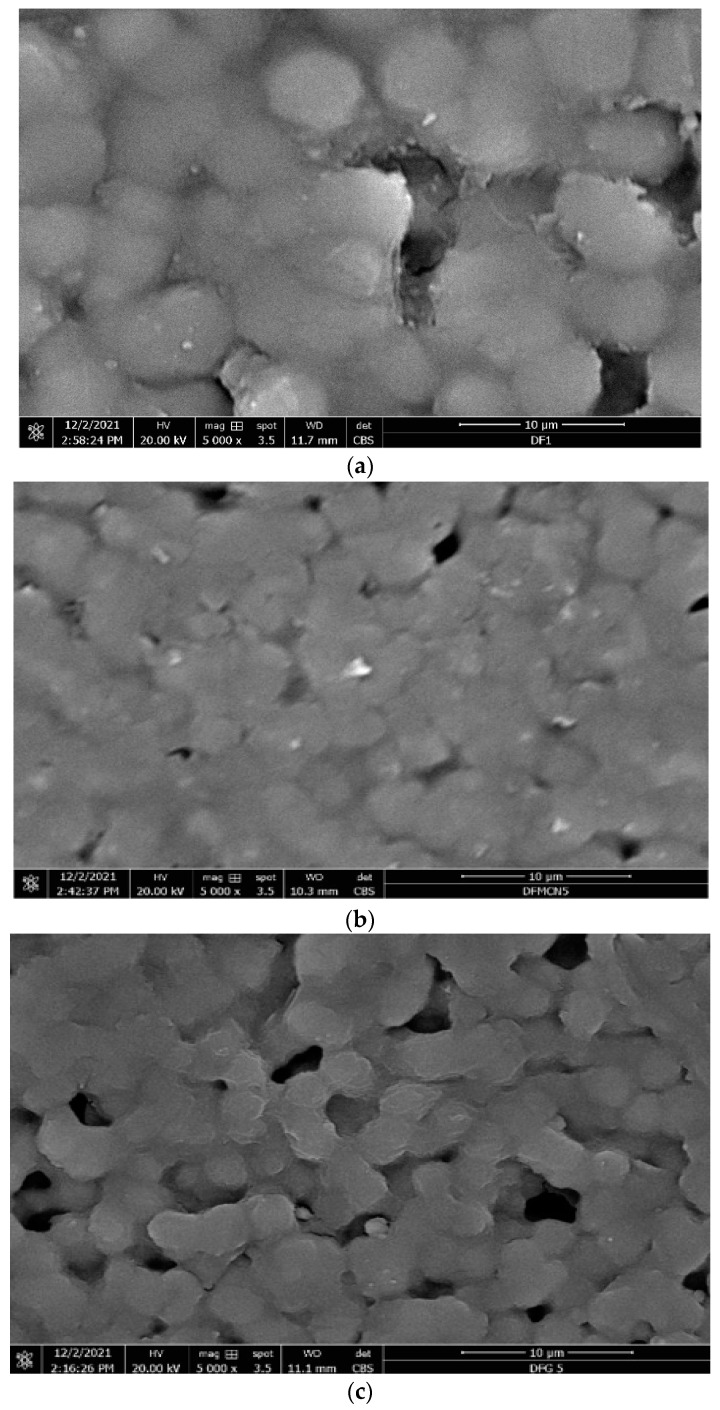
SEM images of (**a**) PVDF, (**b**) MWCNT-PVDF and (**c**) CNF-PVDF nanocomposites.

**Table 1 polymers-14-02976-t001:** Nomenclature of the prepared samples.

Sl. No	Sample Name	Composition
1	DF-1	Virgin PVDF
2	DFG-5-03	PVDF + 5% GNF
3	DFMG-5-02	PVDF + 5% acid-modified GNF
4	DFCN-5-01	PVDF + 5% MWCNT
5	DFMCN-5-01	PVDF + 5% acid-modified MWCNT

**Table 2 polymers-14-02976-t002:** Crystallinity of the nanocomposites obtained from DSC analysis.

Sl. No.	Sample ID	T_c_ (°C)	T_m_ (°C)	% Crystallinity	ΔHf (J/g)
1.	DFCN-5-01	140.04	159.9	37.18	39.09
2.	DFMCN-5-01	141.65	160.37	41.22	43.28
3	DFG-5-03	133.04	159.63	49.19	51.65
4	DFMG-5-02	134.97	159.43	49.79	52.28

**Table 3 polymers-14-02976-t003:** Thermal degradation results of the MWCNT/GNF-PVDF nanocomposites.

Sl. No	Sample Name	T_degrad onset_ Temp °C	T_degrad Peak_ Temp °C	Residue wt.%
1	PVDF	425.00	484.79	20
2	DFG-5-03	458.31	471.31	22
3	DFMG-5-02	461.00	476.5	32
4	DFCN-5-01	428.00	448	24
5	DFMCN-5-01	451.0	471.31	29

**Table 4 polymers-14-02976-t004:** Thermal conductivity of the prepared MWCNT/GNF-PVDF nanocomposite samples.

Sample ID	Average Temperature (°C)	Average Conductivity (W/mK)	
DFG-5-03	22.5	0.1797	0.1853
	32.5	0.1842	0.1894
	42.5	0.1885	0.1930
DFMG-5-02	22.7	0.2585	0.2561
	32.5	0.2609	0.2612
	42.5	0.2640	0.2654
DFCN-5-01	22.5	0.1669	0.1515
	32.5	0.1785	0.1595
	42.5	0.1923	0.1703
DFMCN-5-01	22.5	0.2595	0.2647
	32.5	0.2694	0.2717
	42.5	0.2821	0.2810

**Table 5 polymers-14-02976-t005:** Electrical conductivity of the prepared MWCNT/GNF-PVDF nanocomposite samples.

Sl. No.	Sample ID	Conductivity (S/m)
1.	DFG-0	0.0
2.	DFG-5-3	18.29
3.	DFMG-5-2	35.65
4.	DFCN-5-1	26.82
5.	DFMCN-5-1	26.65

## Data Availability

Not applicable.

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
