# Peer review of "Effect of the Surface Functionalization of Graphene and MWCNT on the Thermodynamic, Mechanical and Electrical Properties of the Graphene/MWCNT-PVDF Nanocomposites"

_polymers, 2022, doi:10.3390/polym14152976_

Round 1

Reviewer 1 Report

The manuscript by Hussain et al. deals with the study of thermo-mechanical and electrical properties of graphene -PVDF and MWCNT-PVDF nanocomposites. Graphene and MWCNT were functionalized with the acid treatment and separately incorporated into PVDF for the fabrication of nanocomposites.  There are some drawbacks that need to be addressed:

  1. “MWCNT/GNF-PVDF nanocomposites” term in the title and manuscript is confusing, there are nanocomposites fabricated from either MWCNT or GNF, Not from MWCNT and GNF together.
  2. The English of the manuscript needs to be improved.
  3. FT-IR Spectra: good quality figures should be provided.
  4. In our previous work….Ref. should be provided.
  5. Line 68, what are “thermo-mechanical dynamic properties”
  6. multiwall carbon nanotube (MWCNT) should be multi-walled carbon nanotube (MWCNT)
  7. 3: Photo of fabricated nanocomposites films should be provided.
  8. Table 1: 4th column for preparation methods can be deleted. (Same as for Tables 4 and 5)
  9. Good quality images should be provided for all figures
  10. “The GNF-PVDF composites showed higher loss modulus values compared to MWCNT-PVDF composites”, explanation desired.
  11. From TGA curves, Figure 8, it is observed that degradation onset for PVDF-CNT nanocomposite is lower than that of pure PVDF, explanation desired.
  12. The interaction between functionalized MWCNT or functionalized GNF and PVDF could be attributed to the interaction between –COOH groups in nanofiller and –Fluorine in PVDF. A similar interaction is represented in Figure 4 in Ref. Nanotechnology 32 (2021) 142004, https://doi.org/10.1088/1361-6528/abcf6c

Author Response

  1. “MWCNT/GNF-PVDF nanocomposites” term in the title and manuscript is confusing, there are nanocomposites fabricated from either MWCNT or GNF, Not from MWCNT and GNF together.

>>> In this manuscript nanocomposites were prepared by adding either graphene or MWCNT, to remove the confusion we rewrite the title as “Effect of the surface functionalization of graphene and MWCNT on the thermodynamic, mechanical and electrical properties of the graphene or MWCNT-PVDF nanocomposites

  1. The English of the manuscript needs to be improved.

>>> We have used English proof reader service in this manuscript

  1. FT-IR Spectra: good quality figures should be provided.

>> Added accordingly

  1. In our previous work….Ref. should be provided.

>> the work is under publication

  1. Line 68, what are “thermo-mechanical dynamic properties”

>> we mean thermal mechanical properties like storage modulus, loss modulus

  1. multiwall carbon nanotube (MWCNT) should be multi-walled carbon nanotube (MWCNT)

>> corrected as multi-walled carbon nanotube (MWCNT)

  1. 3: Photo of fabricated nanocomposites films should be provided.

>> we have added photos in our previous paper thus didn’t added here

  1. Table 1: 4th column for preparation methods can be deleted. (Same as for Tables 4 and 5)

>> deleted accordingly

  1. Good quality images should be provided for all figures

>> modified accordingly

  1. “The GNF-PVDF composites showed higher loss modulus values compared to MWCNT-PVDF composites”, explanation desired.

>> added in line 286

  1. From TGA curves, Figure 8, it is observed that degradation onset for PVDF-CNT nanocomposite is lower than that of pure PVDF, explanation desired.

>> added in line 234

  1. The interaction between functionalized MWCNT or functionalized GNF and PVDF could be attributed to the interaction between –COOH groups in nanofiller and –Fluorine in PVDF. A similar interaction is represented in Figure 4 in Ref. Nanotechnology 32 (2021) 142004, https://doi.org/10.1088/1361-6528/abcf6c

>> Agreed

Reviewer 2 Report

In this manuscript, composite prepared from PVDF with pristine GNF and MWCNT were prepared by the solution casting method. Additionally, the GNF and MWCNT were functionalized by acid treatment and nanocomposites of the acid-treated MWCNT/GNF and PVDF were prepared in the same method. The effect of the acid treatment of MWCNT and GNF on the mechanical, thermal, thermo-oxidative stability, and thermal conductivity of the MWCNT/GNF-PVDF nanocomposites was evaluated.

I consider the content of this manuscript will definitely meet the reading interests of the readers of the Polymers journal. Therefore, I suggest giving a minor revision and the authors need to clarify some issues or supply some more data to enrich the content.

  • For the Keywords, ‘acid treatment’, ‘thermal and mechanical properties’, and ‘solution casting’ should also be added to attract a broader readership and highlight the significance of this work.

  • Line 34, ‘The fabrication of polymer nanocomposites using rigid fillers ... significantly improved mechanical, thermal, structural and electrical properties...In addition to introducing the conclusion that the properties of composites are improved after adding filler, the mechanism behind it should be briefly explained.

For example, 'The dispersion of inorganic nanoparticles into a polymeric matrix gives rise to a hybrid inorganic-organic system where a number of different interactions are established, which strongly affect the physicochemical properties of the host matrix [Electrochimica Acta 309 (2019): 311-325].’

  • Line 40, ‘Poly(vinyl difluoride) (PVDF) has been extensively utilized in different research and application areas because of its excellent stability, high-temperature tolerance and oxidation reactions [8-9].What does it mean by high oxidation reactions? PVDF is very stable and difficult to be oxidized. I consider it should be ‘oxidation resistance/stability’ [Materials 11.12 (2018): 2465ï¼›Journal of Membrane Science 544 (2017): 186-194.].

  • Line 72, ‘ We studied the dynamic mechanical and thermal properties primarily to evaluate the thermo-oxidative stability of the PVDF matrix.No, the research object should be the composite materials (polymer matrix + inorganic particle), not polymer alone.

  • Line 90, ‘The filtered MWCNT/GNF was then washed with acetone to remove the impurities of nanotube side-walls.’For the MWCNT, it is clear to remove the nanotube sidewalls as impurities. But for GNF, what are the impurities? There should be no nanotube sidewalls for the situation of GNF.

  • Line 98, ‘ During the preparation of the composites, a 5 wt.% of MWCNT/GNF with respect to PVDF was taken.Why is only 5wt% selected as the ratio between MWCN/GNF with PVDF? It should be explained better. And why is only DMF selected as the solvent for the solution casting process?

  • Line 142, ‘FTIR spectra of the treated MWCNT and GNF were analyzed using a Perkin Elmer spectrum 1000 FTIR spectrometer The spectra are obtained by averaging how many scans? This information should be provided.

  • Line 145, ‘The surface morphology of the MWCNT/GNF-PVDF composites was analyzed...How about the cross-section morphology? This is very important to demonstrate whether the membrane is homogeneous or bi-layer (polymer-rich phase and filler rich phase, see  Electrochimica Acta 378 (2021): 138133).

  • Line 170, ‘Due to the oxidative treatment with strong acids, defects and smaller fragments were observed in nanotubes.Can the defects and small fragments be observed by FTIR? If not, this should be added to the SEM/TEM part. In addition, for Figure 4, the assignment of peaks can be unreadable. I suggest using high-resolution figures and larger font sizes for the peak assignments. The same applies to Figures 11 to 13.

Author Response

I consider the content of this manuscript will definitely meet the reading interests of the readers of the Polymers journal. Therefore, I suggest giving a minor revision and the authors need to clarify some issues or supply some more data to enrich the content.

  • For the Keywords, ‘acid treatment’, ‘thermal and mechanical properties’, and ‘solution casting’ should also be added to attract a broader readership and highlight the significance of this work.

>> added as suggested

  • Line 34, ‘The fabrication of polymer nanocomposites using rigid fillers ... significantly improved mechanical, thermal, structural and electrical properties...In addition to introducing the conclusion that the properties of composites are improved after adding filler, the mechanism behind it should be briefly explained.

For example, 'The dispersion of inorganic nanoparticles into a polymeric matrix gives rise to a hybrid inorganic-organic system where a number of different interactions are established, which strongly affect the physicochemical properties of the host matrix [Electrochimica Acta 309 (2019): 311-325].’

>> added

  • Line 40, ‘Poly(vinyl difluoride) (PVDF) has been extensively utilized in different research and application areas because of its excellent stability, high-temperature tolerance and oxidation reactions [8-9].What does it mean by high oxidation reactions? PVDF is very stable and difficult to be oxidized. I consider it should be ‘oxidation resistance/stability’ [Materials 11.12 (2018): 2465ï¼›Journal of Membrane Science 544 (2017): 186-194.].

>> corrected

  • Line 72, ‘ We studied the dynamic mechanical and thermal properties primarily to evaluate the thermo-oxidative stability of the PVDF matrix.No, the research object should be the composite materials (polymer matrix + inorganic particle),not polymer alone.

>> Correct, we have corrected the sentence as PVDF composites.

  • Line 90, ‘The filtered MWCNT/GNF was then washed with acetone to remove the impurities of nanotube side-walls .’For the MWCNT, it is clear to remove the nanotube sidewalls as impurities. But for GNF, what are the impurities? There should be no nanotube sidewalls for the situation of GNF.

>> Correct, but there could be extra loose graphene layers

  • Line 98, ‘ During the preparation of the composites, a 5 wt.% of MWCNT/GNF with respect to PVDF was taken.Why is only 5wt% selected as the ratio between MWCN/GNF with PVDF? It should be explained better. And why is only DMF selected as the solvent for the solution casting process?

>> 5% was selected for convenient calculation and to compare with our and others previous work. DMF was found best to dissolve in PVDF.

  • Line 142, ‘FTIR spectra of the treated MWCNT and GNF were analyzed using a Perkin Elmer spectrum 1000 FTIR spectrometerThe spectra are obtained by averaging how many scans? This information should be provided.

>> just one time, multiple running makes more complicated

  • Line 145, ‘The surface morphology of the MWCNT/GNF-PVDF composites was analyzed...How about the cross-section morphology? This is very important to demonstrate whether the membrane is homogeneous or bi-layer(polymer-rich phase and filler rich phase, see  Electrochimica Acta 378 (2021): 138133).

>> we tried but the result was inconclusive.

  • Line 170, ‘Due to the oxidative treatment with strong acids, defects and smaller fragments were observed in nanotubes.Can the defects and small fragments be observed by FTIR? If not, this should be added to the SEM/TEM part. In addition,for Figure 4, the assignment of peaks can be unreadable. I suggest using high-resolution figures and larger font sizes for the peak assignments. The same applies to Figures 11 to 13.

>>>> No fragments can be confirmed FTIR, Figures are adjusted accordingly

Round 2

Reviewer 1 Report

It is observed that the manuscript has not been revised carefully. Further, some of the references indicate similar studies which are not cited and question the novelty of the work. (10.1039/C9RA09459H, 10.1016/j.synthmet.2020.116555)

Therefore, I can not recommend the manuscript for publication in Polymers 

Reviewer 2 Report

The author has responded and revised accordingly based on my previous comments, and I think the current version is more reasonable.